

# Distribution of clones among hosts for the lizard malaria parasite *Plasmodium mexicanum*

Allison T. Neal

Department of Biology, Norwich University, Northfield, VT, United States of America

## ABSTRACT

**Background**. Malaria parasites reproduce asexually, leading to the production of large numbers of genetically identical parasites, here termed a clonal line or clone. Infected hosts may harbor one or more clones, and the number of clones in a host is termed multiplicity of infection (MOI). Understanding the distribution of parasite clones among hosts can shed light on the processes shaping this distribution and is important for modeling MOI. Here, I determine whether the distribution of clones of the lizard malaria parasite *Plasmodium mexicanum* differ significantly from statistical distributions commonly used to model MOI and logical extensions of these models.
**Methods**. The number of clones per infection was assessed using four microsatellite loci with the maximum number of alleles at any one locus used as a simple estimate of MOI for each infection. I fit statistical models (Poisson, negative binomial, zero-inflated models) to data from four individual sites to determine a best fit model. I also simulated the number of alleles per locus using an unbiased estimate of MOI to determine whether the simple (but potentially biased) method I used to estimate MOI influenced model fit.
**Results**. The distribution of clones among hosts at individual sites differed significantly from traditional Poisson and negative binomial distributions, but not from zero-inflated modifications of these distributions. A consistent excess of two-clone infections and shortage of one-clone infections relative to all fit distributions was also observed. Any bias introduced by the simple method for estimating of MOI did not appear to qualitatively alter the results.
**Conclusions**. The statistical distributions used to model MOI are typically zero-truncated; truncating the Poisson or zero-inflated Poisson yield the same distribution, so the reasonable fit of the zero-inflated Poisson to the data suggests that the use of the zero-truncated Poisson in modeling is adequate. The improved fit of zero-inflated distributions relative to standard distributions may suggest that only a portion of the host population is located in areas suitable for transmission even at small sites (<1 ha). Collective transmission of clones and premunition may also contribute to deviations from standard distributions.

Corresponding author
Allison T. Neal, aneal1@norwich.edu

## INTRODUCTION

It is not uncommon for hosts to harbor multiple parasites of the same species. For malaria parasites (*Plasmodium* and related genera), detectable infections always consist of numerous individual parasites. Many of these parasites are produced *via* asexual reproduction from a single precursor cell in the insect vector (zygote/ookinete) and continue asexual replication in the vertebrate host; all of these parasites are genetically identical (or nearly so) and are here referred to collectively as a parasite clone. Vertebrate hosts may also harbor multiple parasite clones, which are thought to be acquired primarily *via* bites from multiple infected insect vectors. The distribution of malaria parasite clones among vertebrate hosts is thus shaped in part by the transmission biology of these parasites. The distribution of clones is of practical interest because interactions among co-infecting clones may influence health-related outcomes such as virulence, transmissibility and resistance to superinfection (*Smith et al. 1999*; *Read & Taylor, 2001*; *Vardo-Zalik & Schall, 2009*).

Modeling plays an important role in our understanding of clone distributions, including for estimating two related parameters: multiplicity of infection (MOI; *e.g.*, *Hill & Babiker, 1995*; *Schneider & Escalante, 2014*) and complexity of infection (COI; *e.g.*, *Galinsky et al., 2015*; *Chang et al., 2017*). Though there is some inconsistency in the use of these terms, MOI often describes the number of parasite clones entering a host, whether or not those clones are distinct (*i.e.*, a host may be bitten by multiple vectors carrying the same parasite clone), while COI describes the number of genetically distinct clones in a host. The distribution of clones among hosts is often modeled using either the Poisson or negative binomial (NB) distribution (*Schneider, 2021*); it is therefore important to know how often these assumed distributions are observed in nature. Additionally, clone distributions may provide insight into the underlying biological processes than generate and maintain these distributions. Below, I summarize the Poison and NB distributions in the context of *Plasmodium* biology and provide three examples of how complexities in the biology of *Plasmodium* biology that are supported by empirical data might lead to alterations to these distributions. This list is not meant to be exhaustive, but merely to demonstrate how the biology of *Plasmodium* can impact the distribution of clones among hosts. Table 1 provides a summary of these points.

### Distributions

Two distributions are commonly used to model count data like the number of infectious bites per host: the Poisson distribution and the NB distribution (*Hilbe, 2014*). These distributions model the number of times an event occurs in a given interval of time or space. When applied to the distribution of malaria clones, the 'events' are the clones themselves (or, equivalently, bites by an infected insect vector) and the units of space are the hosts. In practice, the Poisson and NB distributions used are usually conditional (they are zero-truncated, meaning that they do not include not infected hosts, *Schneider, 2021*), in part to account for the fact that many data sets do not include robust sampling of not infected hosts (*Hill & Babiker, 1995*).

The simpler of the two is the Poisson distribution. The Poisson distribution is described by a single parameter, $\mu$, which is both the mean and variance of the distribution.

**Table 1  Overview of distributions of interest.**

| Biological situation | Predicted distribution |
|---|---|
| Clones are independent; all hosts have equal susceptibility and likelihood of exposure | Poisson |
| Clones are independent; hosts show some variability in susceptibility or exposure that causes clones to clump together | Negative Binomial (NB) |
| Some portion of the population meets criteria above (Poisson or NB); some portion is not exposed | Zero-inflated model (Poisson or NB) |
| Clones are not independent; they are regularly transmitted collectively | Undefined, but likely with an excess of multi-clone infections relative to above distributions |
| Clones are not independent; premunition limits multi-clone infections | Undefined, but likely with a shortage of multi-clone infections relative to the above distributions |

The Poisson distribution has a number of assumptions, including the independence of clones (*i.e.,* having a certain clone does not affect the chances of having another) and the equivalence of the mean and variance (*i.e.,* as the average number of clones per infection increases, so does the variance in clones per infection).

In reality, most parasite distributions are overdispersed relative to the Poisson distribution, meaning that the variance-to-mean ratio is greater than 1. For microparasites like malaria, overdispersion often means that parasite clones show significant clumping or aggregation, with a smaller number of hosts containing a larger number of clones than would be expected if clones were distributed at random. This clumping can occur for a variety of reasons, including if individual hosts vary in their susceptibility or exposure or if parasite propagules are distributed unequally in the environment (*e.g.,* due to variation in vector abundance or vectors carrying multiple clones; *Boulinier, Ives & Danchin, 1996*). The negative binomial (NB) distribution is often a better fit in such cases (*Anderson & May, 1992*). The NB is described by two parameters: the mean, $\mu$, and the dispersion parameter, $\alpha$. When $\alpha$ is 0, the NB is equivalent to the Poisson. Higher values of $\alpha$ indicate greater aggregation (*i.e.,* parasite clones are concentrated in a smaller proportion of the host population). Note that the statistical software R, which is used in this paper, uses $\theta$ (or 'size') in place of $\alpha$. $\theta$ is inversely related to $\alpha$, thus small values of $\theta$ indicate substantial clumping of clones and large values indicate similarity to a Poisson distribution (*Hilbe, 2014*). *Hilbe (2014)* provides useful coverage of both the Poisson and NB distributions.

Deviation from these basic distributions could occur for a number of reasons. For example, clones may not be independent of one another, such as if multiple malaria clones are ingested by a vector and transmitted simultaneously to a new host. *Hill & Babiker (1995)* dismissed this as unlikely, but experimental data suggest not only that multiple clones can transfer to the vector and develop there (*e.g.*, *Huber et al., 1998*; *Vardo-Zalik, 2009*), but also that clones that were underrepresented in the vertebrate may actually tend to increase their relative representation in the vector (*Nwakanma et al., 2008*; *Taylor, Walliker & Read, 1997*; *Vardo-Zalik, 2009*). *Hill & Babiker (1995)* outline an approach to model the
distribution of clones in such a case, but note that we do not have sufficient information about the transmission biology of *Plasmodium* to build a meaningful model at this time. Nonetheless, it seems that this situation would lead to an overabundance of multi-clone infections relative to the Poisson or negative binomial distribution.

Transmission may also be extremely localized such that some portion of an apparently continuous population has little to no chance of contracting the parasite, while another portion is at much higher risk than the overall prevalence of the parasite would otherwise suggest. Although some variation in exposure or susceptibility can generate the clumping and overdispersion characteristic of the negative binomial distribution (*Boulinier, Ives & Danchin, 1996*), major differences would cause deviations. Data suggest that such heterogeneity in exposure is common. Transmission of human malaria is known to show hotspots, where transmission in small areas is much higher than the surrounding region (*Gaudart et al., 2006*; reviewed in *Bousema et al., 2012*). Similar patterns are seen in wildlife parasites. For example, *Eisen & Wright (2001)* measured multiple landscape features at different spatial scales and found that ground cover within 10m of a host's point of capture was the best predictor of infection status with the lizard malaria parasite *Plasmodium mexicanum*.

An extreme case of heterogeneity in parasite exposure in which a portion of the population is not exposed at all and a portion has ordinary variation in exposure can be modeled by a zero-inflated distribution. This distribution assumes some proportion of the population has zero probability of becoming infected while the remainder of the population has clones distributed in accordance with the Poisson or NB distribution. Zero-inflated models are mixed models that include a parameter (the 'zero-probability') describing the proportion of observed 0s (uninfected hosts) that are not a result of the processes that generate the other distribution (Poisson or NB; *Hilbe, 2014*); in other words, some portion of uninfected hosts are not infected because they are not exposed to infectious vectors at all and other uninfected hosts just happen to have not been bitten (but had the same chance of being bitten as the infected hosts). The number of not exposed hosts can be estimated as the zero probability by fitting this zero-inflated distribution. Note that fitting a zero-inflated model should predict a very similar distribution of clones among infected hosts to fitting a zero-truncated (conditional) model, but the former may provide greater insight on whether or not infected hosts are subject to the same underlying forces (*e.g.*, exposure to infected vectors) as infected hosts.

One final example of why standard distributions may not accurately predict clonal distributions relates to intraspecific dynamics within infected hosts; negative interactions among *Plasmodium* clones could lead to a paucity of multi-clone infections. For example, there is ample evidence that established infections can prevent the entry of additional clones, a phenomenon known as premunition (*e.g.*, *Vardo, Kaufhold & Schall, 2007*; *De Roode et al., 2005*; reviewed for *P. falciparum* in *Smith et al. 1999*). Blocking the entrance of additional clones would reduce the number of multi-clone infections because a host might harbor only a single parasite clone even if it had received multiple infectious bites. This might alter the distribution of clones in various ways depending on the transmission intensity and strength of premunition, but it seems likely that it would lead to data that

are underdispersed relative to the Poisson. At the extreme, with very high transmission and perfect premunition (and ignoring the possibility that vectors may transmit multiple clones at once), most hosts would be infected by a single clone; the mean number of clones per infection would be low (near 1 clone per host), but the variance would be even lower.

### Research question

Obtaining reliable data for testing the distribution of clones among hosts can be challenging. One reason is that some parasites (*e.g.*, *Plasmodium falciparum*) are known to sequester in capillaries, making it necessary to take multiple samples from a patient to ensure that sequestered clones are not missed (*Tusting et al., 2014*). Additionally, control efforts like anti-malarial prophylactics (*Tusting et al., 2014*) and vaccine candidates (reviewed in *Arnot, 2002*) can alter MOI. Third, reliable counts of uninfected hosts are not always available due to study design (*e.g.*, uninfected hosts may not be sampled) or risk of false negatives (*Hill & Babiker, 1995*).

I therefore pursue a study of the distribution of clones among hosts using the lizard malaria parasite *Plasmodium mexicanum* in its host the western fence lizard *Sceloporus occidentalis*. This natural host-parasite system has been studied for decades (*Schall & St. Denis, 2013*) and is free from many of the challenges outlined above: the parasite is not known to sequester (*Ford & Schall, 2011*), the hosts are not exposed to chemotherapy or vaccination, and false-negatives are known to be quite rare (*Perkins, Osgood & Schall, 1998*). Additionally, despite differences in biology between *P. mexicanum* and human parasites (*e.g.*, *P. mexicanum* is transmitted by sand flies rather than mosquitoes), many of the biological complexities potentially impacting the distribution of clones among hosts have shown similar patterns between *P. mexicanum* and other malaria parasites (see papers referenced under 'Distributions'). My aim was to determine whether the distribution of clones among hosts differs significantly from the distributions outlined in Table 1.

## MATERIALS & METHODS

### Study sites and sampling

The malaria parasite *Plasmodium mexicanum* infects *Sceloporus occidentalis* lizards in western North America and has been studied since 1978 at the University of California Hopland Research and Extension Center (HREC) near the town of Hopland in Mendocino County California (*Schall & St. Denis, 2013*; *Vardo & Schall, 2007*). Lizards are captured using a fishing pole with attached noose, and a small sample of blood is taken before the lizard is released at its point of capture. Blood samples are used to make thin blood smears for processing with Giemsa stain and a small amount of blood is stored frozen on filter paper (Whatman® qualitative circles or similar) for genetic analysis.

Sampling at HREC is conducted at a number of well-established sites. The parasite shows genetic differentiation among these sites, some of which are only a few hundred meters apart (*Fricke, Vardo-Zalik & Schall, 2010*). Because combining data from multiple sites or years might cause deviations from standard statistical distributions, I tested the fit of the selected distributions using data taken from the four individual sites sampled in 2009 or 2010 that had the most infections: Water Tank (WT, 2009), Parson's Creek (PC,

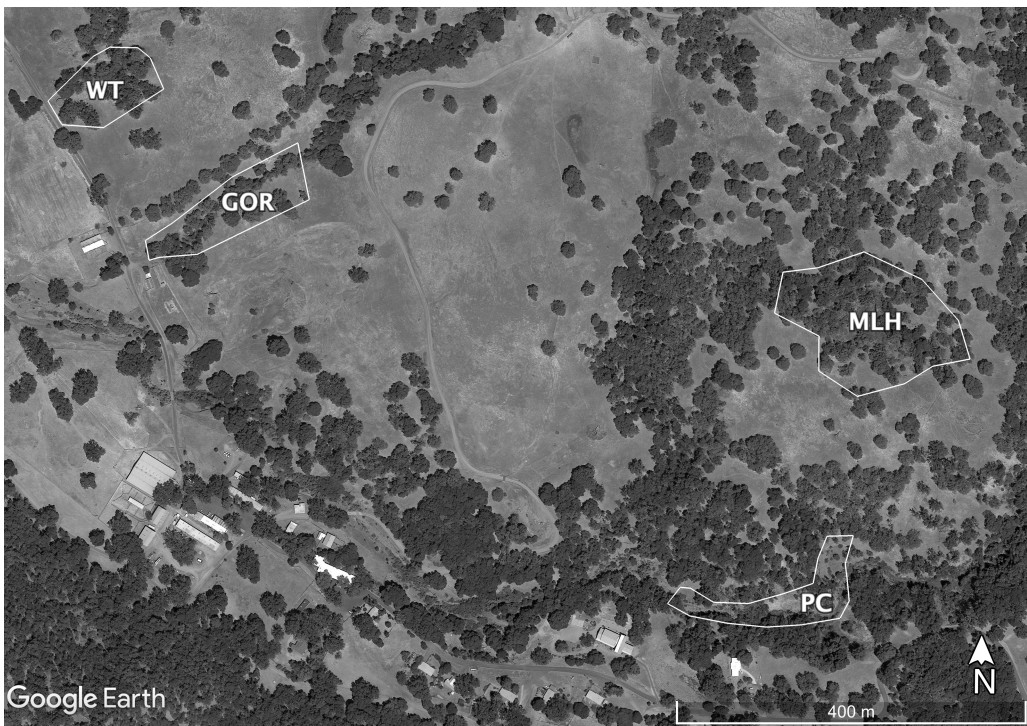

**Figure 1** **Map of the sites sampled for this study.** Sites are Gorge (GOR), Middle Lower Horse (MLH), Parson's Creek (PC) and Water Tank (WT). Map data ©2021 Google.

2009), Middle Lower Horse (MLH, 2010) and Gorge (GOR, 2010). The largest of these sites, MLH, is less than 4.5 ha; *Eisen & Wright (2001)* describe this site in detail, though the area sampled in 2009–2010 is a bit smaller than their study area. The other sites are each closer to 1ha (Fig. 1).

*Plasmodium* infections in this system are known to be chronic; as a result, variation in the duration of exposure may also cause deviations from standard distributions (*i.e.,* older lizards might harbor more parasite clones because they have been exposed to infectious bites over a longer period). To limit the impact of variation in lizard age on clonal distributions, I retained only lizards in a limited size range (40–64 mm) in the final analysis. For lizards collected in early summer (the vast majority of lizards in this data set), this range of sizes corresponds to lizards that hatched in late summer of the previous year with a high level of confidence (*Davis, 1967*).

## Genetic analysis

All infections included in this study were genotyped previously; these data were published by *Neal & Schall (2014)*. Genotyping was performed for 4 microsatellite loci- Pmx306, Pmx732, Pmx747 and Pmx839- using fluorescently labeled PCR primers and conditions described by *Schall & Vardo (2007)*. The PCR product was diluted and underwent fragment analysis at Cornell University Biotechnology Resource Center.

The parasite is haploid in vertebrate blood, so each clone will have a single allele at each locus and produce a single peak on an electropherogram. Clones that are at a low relative frequency in the blood may be hard to detect in infections due to a weak signal on the electropherograms. No peak-size cut-off was applied to correct for this, but a previous study demonstrated that even alleles with 10-fold differences in concentration could usually be detected (*Vardo-Zalik, Ford & Schall, 2009*). Additionally, alleles were classified by relative electropherogram peak height (highest, >1/3 highest, <1/3 highest [small]); these designations were not used in my analysis, but are available in the data set at osf.io/6yvh5.

I estimated the number of clones per host as the maximum number of alleles seen at any one locus (a common measure also used for human malaria (*Zhong et al., 2018*)). Loci with fewer alleles may result if clones share alleles at that locus. This measure provides a minimum estimate of the number of clones per infection because some clones may share alleles, and will be referred to hereafter as the observed minimum number of clones, or observed MC. *Anderson et al. (2010)* showed that it was possible to distinguish about 90% of *Plasmodium falciparum* clones using only four microsatellites, and the loci in that paper had lower average heterozygosity than the loci I used. This suggests that observed MC, while a minimum estimate of MOI, may not greatly underestimate the number of distinct clones in most cases.

## Fitting distributions

All analyses were performed in R (R Development Core Team, Vienna, Austria). The following analyses were performed on observed MC data for each of the four single sites individually (GOR [2010], MLH [2010], PC [2009] and WT [2009]). I fit the Poisson distribution using the glm() function and the NB using the glm.nb() function. The zero-inflated models were fit using the zeroinfl() from the R package "pscl". I assessed dispersion using the Pearson dispersion statistic as suggested by *Hilbe (2014)* and compared each zero-inflated model to the corresponding standard model with the Vuong test using the function vuong(). I performed a chi-squared test comparing the observed *vs.* expected values of each distribution after grouping multi-clone infections to reduce the number of categories with fewer than five observed infections following (generally) the suggestion of *Bliss & Fisher (1953*; two sites unfortunately had fewer than five single-clone infections: PC had four and WT had three). Due to the relatively small number of infections at each site considered individually, I used Fisher's combined probability test to obtain an overall assessment of whether the distribution of observed MC at the four sites collectively differed significantly from each distribution.

As mentioned above, observed MC is known to provide a minimum (and therefore biased) estimate of the true MOI. True MOI is almost certainly higher because some distinct clones will share alleles and some hosts may have been infected by the same clone multiple times (*Schneider, 2018*). If the bias is slight, it may not greatly affect the fit of distributions, but if the bias is greater, it could cause MOI estimates to deviate from expected distributions even if true MOI does not. To account for this possibility, I used the likelihood approach of *Schneider & Escalante (2014)* to obtain an unbiased

(*Schneider, 2018*) estimate of average MOI at each site, simulated the observable number of alleles per locus using a Poisson distribution of clones per infection and allele frequencies observed at each site, and compared this expected distribution to the distribution of observed MC. True MOI and true allele frequencies were estimated from genetic data for each site using the MLE() R function provided by *Schneider (2018*; supplemental materials). A separate MOI estimate was obtained for each of the four microsatellite loci, and these values were averaged to obtain a single overall estimate of MOI for each site. I then simulated the distribution of clones among $n_s$ hosts, where $n_s$ is the number of lizards collected for the observed data from each site $s$. Simulations were performed by drawing $n_s$ random numbers from a Poisson distribution with lambda equal to the estimated MOI. These random numbers represent the number of infectious bites received by each of the $n_s$ simulated hosts. I then used the function mnom(), also from the R code in *Schneider (2018*; supplemental materials), to randomly draw $x_i$ alleles from a multinomial distribution with allele frequencies estimated from the observed data (described above), where $x_i$ is the random number of infectious bites for each host drawn from the Poisson distribution in the previous step. Sampling of the multinomial distribution was performed for each of the $n_s$ hosts at each of the four microsatellite loci. After allele sampling, a count of the number of distinct alleles per locus was tallied (note that some alleles may be sampled more than once, but would only be counted once) and the maximum number of distinct alleles at any locus was recorded as simulated MC. Taken together, these steps were meant to simulate the expected underlying biology (Poisson distribution of infectious bites) and known observational limitations (overlap in alleles among clones, repeated infection with the same clones) of the observed data. The simulation was repeated 1,000 times to determine the mean, minimum and maximum number of infections observed with each possible number of clones to determine if observed data fell in this range (estimation of a 95% confidence interval did not seem necessary; see Results).

## RESULTS

Forty-five (WT) to 131 (MLH) lizards were sampled per site, with the number of infections per sampled site ranging from 10 (PC) to 17 (MLH; Table 2). Despite these relatively small sample sizes, there was still sufficient genetic variation to differentiate clones; the number of alleles per locus at individual sites ranged from 5 to 10, with every site having at least one locus with 7 or more alleles (See Table S1 for details).

The distribution of observed MC was overdispersed relative to the Poisson (dispersion parameter range 1.55 [WT]–2.12 [GOR]) and differed significantly from this distribution (combined $P < 0.0001$, $X^2 = 112.5$, $df = 8$). The distribution also differed significantly from the NB (combined $P = 0.0031$, $X^2 = 23.2$, $df = 8$), showing significant underdispersion relative to the NB (dispersion parameter range 0.75 [WT]–0.80 [MLH]). The fit values of theta for the ZINB were so large (all > 20,000; Table S2) that the fit distributions were indistinguishable from the ZIP, and the observed data did not differ from either of these fit distributions (combined $P = 0.296$, $X^2 = 9.58$, $df = 8$; values are for both ZIP and ZINB). Table 2 shows the fit values and goodness-of-fit statistics for the
**Table 2  Summary and fit statistics for sites used to test the distribution of clones among hosts.**

| Site | Year | $n_s$ | inf | Negative binomial | | | | ZI Poisson | | | |
|------|------|-------|-----|------|-------|------|--------|------|------|-------|------|
| | | | | *mu* | *theta* | chisq | P | *mu* | *zp* | chisq | P |
| GOR | 2010 | 77 | 12 | 0.30 | 0.17 | 0.73 | 0.70 | 1.48 | 0.80 | 0.085 | 0.96 |
| MLH | 2010 | 131 | 17 | 0.22 | 0.17 | 3.48 | 0.18 | 1.18 | 0.81 | 0.833 | 0.66 |
| PC | 2009 | 78 | 10 | 0.23 | 0.15 | 11.91 | **0.0026** | 1.31 | 0.82 | 5.96 | 0.051 |
| WT | 2009 | 45 | 14 | 0.60 | 0.57 | 7.1 | **0.029** | 1.49 | 0.60 | 2.70 | 0.26 |

**Notes.**
Site indicates where lizards were collected (Gorge [GOR], Middle Lower Horse [MLH], Parson's Creek [PC], and Water Tank [WT]). Year indicates the year samples were collected. For each site and year, I report: $n_s$- the number of lizards collected that were 40–64 mm snout to vent; inf- the number of these lizards that were infected with *P. mexicanum* and have genetic data (only one lizard from GOR was infected but not genotyped); mu- mean clones per host from fit negative binomial or zero inflated Poisson distributions theta- dispersion parameter of clones per host from fit negative binomial distribution; chisq- chi squared value from goodness of fit test (observed *vs.* expected from fit distribution); P- P-value from goodness of fit test (observed vs. expected from fit distribution); zp- the zero probability from the fit zero inflated Poisson distribution. Two additional models were fit, with results reported in the supplemental materials. The model selected for presentation here are the best fitting traditional model (Negative binomial) and zero inflated model (ZI Poisson).

NB and ZIP. Table S2 shows individual goodness-of-fit $X^2$ tests for each site individually, and Table S3 lists the fit values for each of the four models for every site. AIC and BIC are both lower for the traditional NB than the traditional Poisson at every site (Table S4). The Vuong Test indicates that the ZIP models are preferable to the traditional Poisson models ($P < 0.05$ for all raw, AIC- and BIC-corrected tests; Table S5) and that the ZINB models may be indistinguishable from or slightly better than the traditional NB ($P$-values mixed; Table S5). A comparison of the observed and expected values for all distributions at all four sites is shown in Fig. 2, and the amount each model deviated from observed values is shown in Figs. 3A–3C.

The distribution of simulated MC differed considerably from observed MC. Most notably, the simulation substantially overestimated the number of hosts that would be infected: estimates ranged from 32 (WT) to 87 (MLH) infections per site, while observed numbers of infections were much lower (10 [PC] to 17 [MLH]). For all four sites, the observed numbers of uninfected lizards were well above the range observed in 1000 simulations and the observed numbers of lizards with a single apparent clone (MC = 1) were below the range (Table S6). Looking just at the distribution of clones among infected hosts, values of observed MC showed a consistent deficit of single-clone infections and excess of two-clone infections compared with the relative frequencies of these infection types in the simulation (Fig. 3D), much like the distributions fit to observed MC (Figs. 3A–3C).

## DISCUSSION

Many models of MOI for malaria parasites assume that *Plasmodium* clones are distributed among hosts following the Poisson or NB distribution, but certain aspects of these parasites' biology may cause substantial deviations from this prediction (*e.g.*, Table 1). I compared the observed numbers of clones per infection at four individual sites with those expected based on four statistical distributions: the traditional Poisson, the traditional NB, the zero-inflated Poisson (ZIP) and the zero-inflated negative binomial (ZINB). The observed data differed significantly from both traditional distributions, but do not differ from either

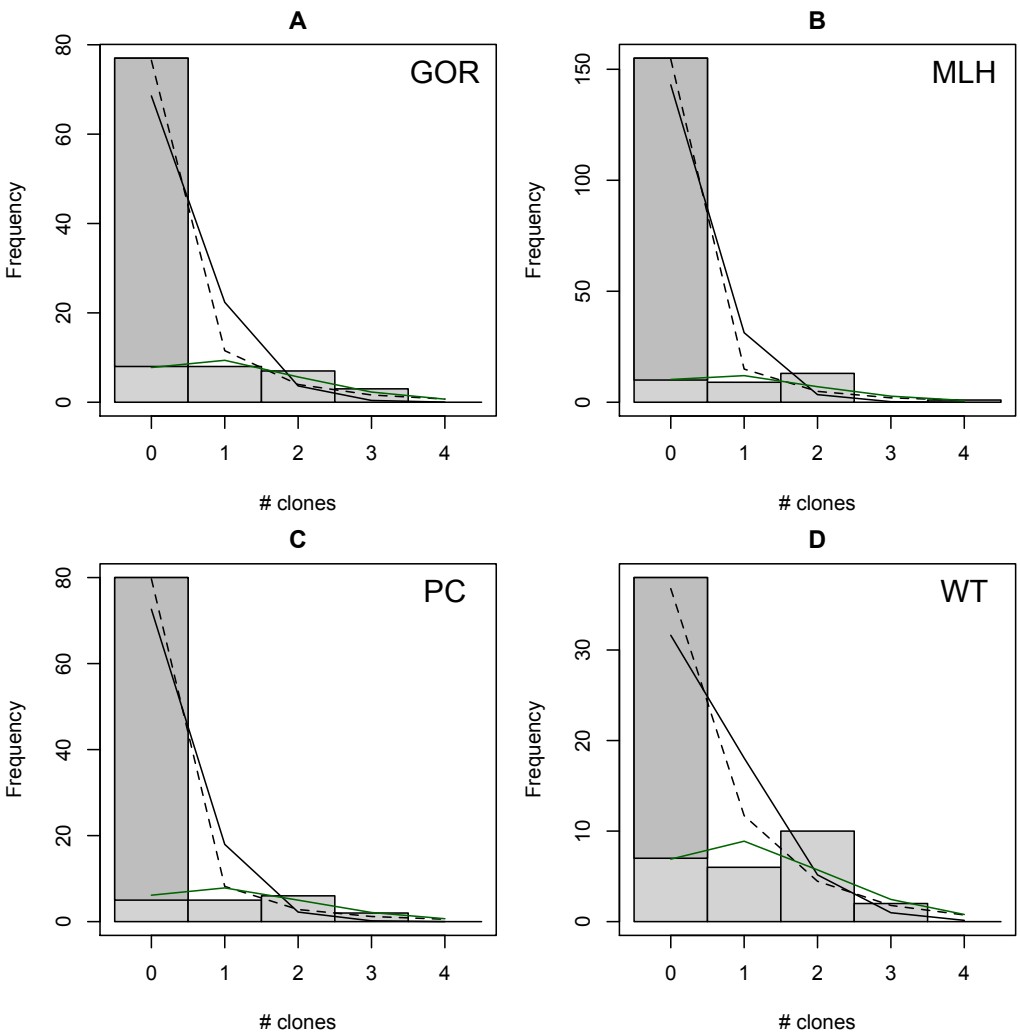

**Figure 2** **Observed (bars) *vs.* expected (lines) number of parasite clones per host at four individual sites (A–D).** Expected lines are based on traditional Poisson (solid black), traditional negative binomial (dotted black), and zero-inflated Poisson (solid green) models (the zero-inflated negative binomial fit is equivalent). The full height of the first bar (0 clones) represents the observed number of not infected lizards at the site; the dark gray portion of the bar represents the number of lizards estimated to be not exposed based on the fit zero-inflated distribution (the estimated zero probability). Collectively, the distribution of clones among infections differs significantly from both traditional models but not the zero-inflated models (see text for details).

zero-inflated distribution. The fit values of theta for the ZINB were so large (all >50,000) that this model essentially converges on the ZIP distribution (recall from the introduction that large values of theta indicate similarity to the Poisson). While the method used to measure MOI in this study is expected to show bias (discussed above), incorporating an unbiased estimate of MOI (simulation) did not appear to qualitatively alter the results: the Poison distribution still did a poor job of describing the distribution of clones among hosts when not infected hosts were considered (Table S6), and still appeared to overestimate the relative frequency of one-clone infections and underestimate the relative frequency of

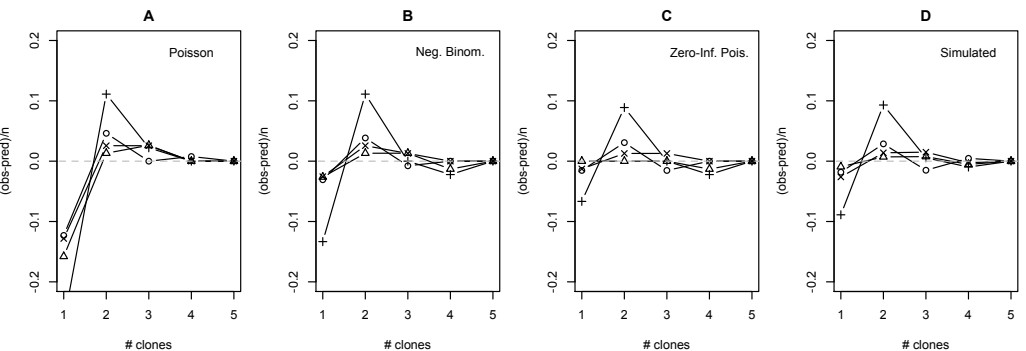

**Figure 3  Deviation from fit distributions of number of *Plasmodium mexicanum* clones per infected host.** Fit distributions are (A) Poisson, (B) Negative binomial, (C) Zero-Inflated Poisson (Zero-Inflated Negative Binomial is equivalent) and (D) Simulated distribution based on MOI estimated using the method of *Schneider & Escalante (2014)*. Data are shown for four individual sites: GOR (triangles), MLH (circles), PC ('X'), and WT (+). Because MOI was estimated from infected lizards only, simulated expectations were scaled down to equalize the total number of expected *vs.* observed infections; without this adjustment most simulated values were well above observed values.

two-clone infections among infected lizards (Fig. 3). Taken together, I conclude that the best fitting distribution for the observed data is the ZIP.

The ZIP was selected for comparison with data because previous studies have shown that malaria transmission can be highly localized. The fact that this distribution showed the best fit of those tested may suggest that localized transmission is important in shaping the distribution of clones among hosts. In fact, the fit value of one of the parameters, the zero probability, suggests that a high percentage (60–82%) of lizards may have essentially no chance of contracting malaria even at sites where *P. mexicanum* is relatively common. In other words, sites may contain small patches where most transmission is focused surrounded by areas where transmission is much lower or non-existent. The feasibility of such hotspots being established is facilitated by the fact that *S. occidentalis* lizards show high site fidelity, often being found on the same rock, log or fence post throughout the collecting season (A Neal, 2010 pers. obs.; *Eisen & Wright, 2001*), and that the insect vectors (*Lutzomyia* sand flies) are not strong fliers (*De Queiroz Balbino et al., 2006*). Together, this sets up a situation in which transmission could be highly localized, and previous research has shown that certain landscape features (*e.g.*, leaf litter) are associated with higher rates of infection with *P. mexicanum* on a scale of a few meters (*Eisen & Wright, 2001*).

Even though the ZIP was the best fitting model, there were still consistent discrepancies between the model's predictions and the observed data. Most notably, all four sites showed a consistent shortage of single-clone infections and a consistent excess of two-clone infections relative to model predictions (Fig. 3), regardless of the distribution examined. I have no definitive explanation for this pattern, but I discuss a few possibilities below.

One possibility is that there is a consistent bias in the way that we estimate the number of clones per infection. Estimates of MOI based on the maximum number of microsatellite alleles at a locus do have known limitations. For one, microsatellite data itself is imperfect. PCR may preferentially amplify more common or shorter fragments, and minor clones

may be missed due to the relative weakness of the electropherogram signal, especially if the minor clone carries an allele that falls in the stutter (a common artifact in microsatellite analysis) of a major clone's allele. While this remains a legitimate concern, previous research conducted on *P. mexicanum* using the microsatellite loci reported on here suggest that, while not perfect, genotyping results with these markers are fairly robust even with 10-fold differences in the relative proportions of clones (*Vardo-Zalik, Ford & Schall, 2009*). A related concern is that *Plasmodium* clones may share alleles at a locus or experience superinfection from the same parasite clone, could make true true MOI higher than observed MC. Although overlapping alleles would tend to underestimate the number of clones found in an infection, possibly leading infections with three or more clones to be misclassified as two-clone infections, it does not seem likely that this explains excess of two-clone infections in the data because the same pattern was seen when comparing observed data to the MLE simulation, which incorporates an unbiased estimate of MOI.

Another possibility is that the consistent excess of two-clone infections reflects something biologically important. Two of the biological situations outlined in Table 1 were not connected with specific distributions to be tested in this study, but I instead predicted that they would lead to an overabundance (collective transmission) or underabundance (premunition) of multi-clone infections relative to the fit distributions. What is observed in the data is a consistent excess of multi-clone infections, but only those with two clones. This does not lead to a clear support of either scenario in isolation, but could conceivably result from the two forces combined. It is possible that small numbers of clones (*e.g.*, two) are transmitted collectively and, once established, are resistant to superinfection (*i.e.,* display premunition), perhaps more so than single-clone infections. Data from *P. falciparum* do suggest that complex infections are more likely to display premunition (*Smith et al. 1999*). It does not appear that is the case for *P. mexicanum* (*Vardo, Kaufhold & Schall, 2007*), though single and two-clone infections were grouped for analysis in that experiment.

## CONCLUSIONS

The data presented here do not provide any definitive answers about what shapes the distribution of *Plasmodium* clones among hosts- that was never my intention. The observed distribution of clones may follow or diverge from standard statistical distributions for any number of reasons. What these data do provide is insight into whether commonly held assumptions about these distributions are supported by data and guidance for shaping future hypotheses.

First, the data presented here provide support for the use of the conditional Poisson in modeling MOI and the distribution of clones among hosts (as is used, for example, in *Schneider & Escalante, 2014*; *Hill & Babiker, 1995*; *Schneider, 2021*). Zero-truncated distributions are conditional distributions that leave out uninfected individuals and have frequently been used in these models. Their use was motivated primarily by limitations in data (*e.g.,* uninfected individuals are not sampled or may be misclassified due to low parasite density in the blood), but conveniently, these zero-truncated distributions may also provide a good fit for data that would reasonably be described by zero-inflated

distributions, which could result from extremely patchy transmission. My data suggest that zero-inflated models provide a better fit to MOI data for *P. mexicanum* than traditional models, and zero-truncated models should therefore also do an adequate job of modeling these data.

Second, this study suggests that transmission hotspots may be as important for the transmission of *P. mexicanum* as they are for human malaria parasites. Highly localized transmission may account for the improved fit of zero-inflated models relative to traditional distributions. Additional research into fine-scale transmission patterns in both human and animal parasites may prove fruitful.

Finally, the distribution of clones among hosts at the individual sites sampled here appear to show a consistent excess of two-clone infections and shortage of single-clone infections relative to the predictions of any of the fit distributions. I am at a loss to provide a clear hypothesis regarding the source of this deviation, but it may be interesting to determine whether such patterns are seen for other malaria parasites and an explanation for this pattern, when found, may provide new insight into the biology of these organisms.

## ACKNOWLEDGEMENTS

I am incredibly grateful to the six anonymous reviewers who provided invaluable suggestions at various stages in the development of this manuscript. I also thank J Schall and his students for helping curate the data set analyzed here and the staff of the University of California Hopland Research and Extension Center for their continuing support of this research.

### Funding

All data reported in this paper were collected for previous studies. Funding for those studies may be found in the corresponding papers (see Methods section for references). Allison T. Neal was supported by a United States National Science Foundation Graduate Research Fellowship during the early stages of this project. The funders had no role in study design, data collection and analysis, decision to publish, or preparation of the manuscript.

### Grant Disclosures

The following grant information was disclosed by the author:
United States National Science Foundation Graduate Research Fellowship.

### Competing Interests

The author declares there are no competing interests.

### Author Contributions

- Allison T. Neal conceived and designed the experiments, performed the experiments, analyzed the data, prepared figures and/or tables, authored or reviewed drafts of the paper, and approved the final draft.

## Data Availability

The data and R code are available at OSF: Neal, Allison T. 2021. "Distribution of Plasmodium Mexicanum Clones among Hosts." OSF. June 16. https://osf.io/6yvh5/.

## Supplemental Information

Supplemental information for this article can be found online at http://dx.doi.org/10.7717/peerj.12448#supplemental-information.

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
