# Peer review of "Distribution of clones among hosts for the lizard malaria parasite Plasmodium mexicanum"

_PeerJ, doi:10.7717/peerj.12448_

## Round 0.1 · original submission · Major Revisions

· Academic Editor

Major Revisions

The review process is now complete, and two thorough reviews from highly qualified referees are included at the bottom of this letter. Although there is considerable merit in your paper, we also identified some concerns that must be considered in your resubmission. I strongly agree with them and emphasize that the author must dedicate to answer the points raised with the utmost precision and to make all those reconsiderations in the manuscript to be submitted.

Reviewer 1 ·

Basic reporting

Clarity of the manuscript. This is an interesting article, unfortunately, the strengths of the manuscript are currently obscured by the writing style. The author has often used long sentences, and sometime hard to follow, as many unanswered questions discussed. I think that the same information could be presented in a more condensed form. The manuscripts reads as a graduate thesis, with lots of redundancy.

Background. The author has presented an elaborative background covering most of the augments on the MOI. However, the majority of cited data are conventional and relatively old, with exception one published in 2018, all references are almost a decade old.
There is a lack of references to some approaches addressing factors influencing the distribution of malaria parasites in nature. For example, a large body of literature on the human malaria parasites P. falciparum and P. vivax have shown spatial heterogeneity has a profound effect on predictions of MOI and parasite prevalence. The author has not considered these findings, to argued the approach taken in the present study.

Structure. The article is prepared in standard format and relevant data is presented in a well analysed forms in tables and figures, adequately addressing the research question making a coherent story - supported by appropriate data. However, the elaborative writing style, makes some sections read redundant. Thus the author should may consider shortening some sections (e.g introduction and discussion).
Figures. The figures are clear and well labelled containing essential data addressing the research question.
Raw data. It would be of great help to interested readers to have the microsatellites data for the examined isolates

Self-contained with relevant results. The manuscript contains appropriate material and results addressing the research question.

Experimental design

Research question. Aim and research question are well defined, and the study design and approach used suit the research question. However, as mentioned above, I am not sure how novel is the research question?
Association between MOI and parasite prevalence (surrogate measure of transmission) has been addressed extensively, many studies have reported a clear association among natural infection of the human malaria parasites (P. falciparum and P. vivax) (as mentioned in the intro). Nonetheless, the relationship between MOI and parasite prevalence is not simple, this has been attributed to many epidemiological and biological factors (as stated by the author). Many studies have clearly demonstrated the impact of spatial crusting of transmission on relationship between MOI and parasite prevalence (e.g. Bousema T, et al., PLoS medicine; 9(1):e1001165). Karl Sl;, PLOS ONE DOI:10.1371). Heterogeneity is maintained in all transmission setting, its not easy to identify in high transmission settings. The author has not considered this point in the study design, and has not adequately bring it in the discussion.
The author has listed challenges to estimates of MOI in nature, however it is not clear how the design of the current study overcomes the above limitations.
Analysis of distribution of P. mexicanum clones is interesting, and adds to existing models for analysis of MOI. This the manuscript, should be restructured to make the model of MOI as a major theme. The titles should be amended to reflect the main focus of the manuscript.

Investigation performed and methods. The methodology section contains adequate information on data collection and analysis. However, there it is not clear if the author has used existing genotype data for the examined microsatellites or whether the author has generated this data. Thus, some restructuring can help, e.g. sub-section on “microsatellites analysis”. Line 263. All infections in this study that had available dried, frozen blood dots? Are the samples stored dried on filter papers? What type? Has the long term storage effect on MOI been verified? Any cut-off peak height been applied for detection of multiple clones per locus, to remove spurious fragments, to follow on this, Line 420 - 22 “the minor clone may be missed due to the relative weakness of the electrophoresis signal, ….that falls in the stutter” . These sentences are not clear.
In addition, the sub-section “Individual sites for testing goodness of fit”, can be added to the analysis?
There is some overlap, lines 229-33 in section “MOI at high vs. low prevalence sites” better fit in section “MOI at high and low prevalence”.
The analysis and interpretation of the data are adequate,
Line 339, S3 table has come prior to S2, rearrange the order of table S3 and S2.
As mentioned above, It would be of help to present the row, on microsatellites and parasite isolates analysed.

The discussion is limited in scope; the author has adequately compared her data to previous data on P. maxicanum and some studies of the human malaria parasites. However, limitations in approach were not adequately presented.

Validity of the findings

The author has presented an extensive data, complied over many years, summarized/analysed in clear tables and figures in line with the research question. However, I am not sure about the novelty of the research question. It would therefore be of benefit to clarify how the approach taken in the present study extends existing body of knowledge.
The above is reflected in the conclusion, line 456 “What these data do provide is guidance for
shaping future hypotheses and insight into whether commonly held assumptions about MOI are supported by data”, line 461 “The use of MOI as a proxy for transmission or prevalence (or vice versa) warrants substantial caution” and “Third, this study provides a reasonable hypothesis for why high vs. low prevalence sites may not differ in MOI”

The 1st para of conclusion can easily be taken out without affecting the main points raised in this section.

Lines 478-85. The third point in the conclusion. There are many speculations here, parasite prevalence is not a sensitive epidemiological index, EIR provides a more meaningful transmission intensity index. Many previous work has established that MOI is a nonlinear function of parasite prevalence.

Reviewer 2 ·

Basic reporting

The data can be presented in a more readable way. It is not clear how close were the sampled locations, representing them on a map would be very helpful. In addition, I suggest further describing how the low prevalence locations were grouped and what were the ranges defining low versus high prevalence. Is figure 1 showing the distribution by pooling the data across all sites? How do the fitted distributions look for each side individually?

To ensure reproducibility of the study, the code and the data should be provided as well.

Experimental design

While the research question that the manuscript addresses is of great importance, this question has been addressed in many previous studies and I am not convinced that the data and analysis presented in this manuscript stands to the conclusions that the author wants to draw. I think that by looking at the MOI in lizard populations, once cannot draw general conclusions about malaria in human hosts. There is a fundamental difference in the disease biology in humans compared to lizards, not to mention the different vectors, the intervention effects and other aspects of human malaria which affects this relationship. Furthermore, it is widely accepted that transmission intensity is linked to MOI, especially at large spatial scales and here are just a few studies that have shown that:
https://pubmed.ncbi.nlm.nih.gov/29325146/
https://www.ncbi.nlm.nih.gov/pmc/articles/PMC3537863/

Validity of the findings

While I agree that the relationship between MOI and prevalence is complex and should be further investigated, the author tries to claim that MOI and transmission intensity are not linked just because the analysis of one particular data set did not support it. Drawing these types of conclusions and extrapolations need to be done with particular caution. I suggest a rewrite of the manuscript, focusing on the presented data and the results, without extrapolating these results to human malaria.

Additional comments

The manuscript focuses on dissecting the relationship between the observed multiplicity of infection (MOI) and inherent malaria transmission. To address this matter, the author studies the MOI from historical samples collected between 1995-2010 from lizard hosts infected with Plasmodium mexicanum and the relationship of the MOI with the observed prevalence in the lizard populations sampled across several locations in California between 1980-2010. To determine the MOI, the observed allele diversity at 4 microsatellite loci is assessed.

While the research question that the manuscript addresses is of great importance, this question has been addressed in many previous studies and I am not convinced that the data and analysis presented in this manuscript stands to the conclusions that the author wants to draw. I think that by looking at the MOI in lizard populations, once cannot draw general conclusions about malaria in human hosts. There is a fundamental difference in the disease biology in humans compared to lizards, not to mention the different vectors, the intervention effects and other aspects of human malaria which affects this relationship. Furthermore, it is widely accepted that transmission intensity is linked to MOI, especially at large spatial scales and here are just a few studies that have shown that:
https://pubmed.ncbi.nlm.nih.gov/29325146/
https://www.ncbi.nlm.nih.gov/pmc/articles/PMC3537863/

While I agree that the relationship between MOI and prevalence is complex and should be further investigated, the author tries to claim that MOI and transmission intensity are not linked just because the analysis of one particular data set did not support it. Drawing these types of conclusions and extrapolations need to be done with particular caution. I suggest a rewrite of the manuscript, focusing on the presented data and the results, without extrapolating these results to human malaria.

Furthermore, the way the statistical analysis of the data is done and presented needs further improvement. While Poisson and Negative Binomial distributions are often used to represent count data, and the assumptions of many models, they are not a "gold standard" and the observed data does not necessarily have to follow them. It is unclear why the author emphasizes that these distributions are expected to reproduce the data. For example, I would have considered power law distributions which are more widely used for describing genomic properties.

An important limitation of the study: the author assumes that MOI can be estimated by taking the maximum number of observed alleles at any of the 4 microsatellite loci. This number is a lower bound of the true MOI and thus it is not surprising that there is little to no difference between low and high prevalence sites. A more robust measure of MOI, potentially incorporating the variation observed at all the 4 loci would be more adequate here.

The data can be presented in a more readable way. It is not clear how close were the sampled locations, representing them on a map would be very helpful. In addition, I suggest further describing how the low prevalence locations were grouped and what were the ranges defining low versus high prevalence. Is figure 1 showing the distribution by pooling the data across all sites? How do the fitted distributions look for each side individually?

To ensure reproducibility of the study, the code and the data should be provided as well.

---

## Round 0.2 · Major Revisions

· Academic Editor

Major Revisions

Your manuscript went through another round of reviewers who carefully analyzed the revised version and pointed out several flaws that still deserve attention. As you can see, both reviewers highlight the need to clarify the method used and the resulting conclusions. Please, provide, point-to-point responses according to the comments made the Reviewer #3 and #4 in the new version of your manuscript.

Reviewer 2 ·

Basic reporting

no comment

Experimental design

no comment

Validity of the findings

no comment

Additional comments

Thank you addressing my review comments.

Reviewer 3 ·

Basic reporting

Exposition is clear and concise

Experimental design

The methods and experimental design are relevant

Validity of the findings

I have many reservations about findings, explained in my letter

Additional comments

The paper aims to assess distribution patterns of MOI (multiplicity of infection) using field data for lizard malaria, and 4-locus microsatellite analysis.
The issue of malaria MOI distribution and its relation to transmission (intensity and prevalence) is wrought with complexities and controversies, brought up by the author and the reviewers. Just quantifying diversity poses a challenge; the proposed 4-locus microsatellite could be a reasonable approach. The next big question is what MOI pattern one should expect, and its relation to transmission intensity (EIR) and parasite diversity in host population.
Beyond qualitative inference (‘higher EIR implies higher MOI’ and prevalence), I don’t know any direct way (mathematical or other) to assess it. Fitting data by itself has limited value, in my view without an explanation.
In that sense, MOI could be contrasted to another commonly used ‘Poisson’ / ‘Negative binomial’ distribution describing parasite burden in population. Those could be derived by conventional statistical (event-count) methods, mentioned by the author. However, burden distribution does not translate directly to MOI, for multiple reasons, some mentioned by the author. I also have reservation about lack of correlation between ‘infection prevalence’ and MOI, and what conclusions to draw. Besides, ‘prevalence’ has limited values in the context of malaria.
Overall, I share the previous comments, and defer it to their judgement.

Reviewer 4 ·

Basic reporting

The manuscript is overall well written and the quality of the language is sufficient. However, parts of the methods section are difficult to understand and seem contradictory at first read. I had to go over it several times to understand the methodology. This can be improved by adding more structure with subheading to the methods section.


The references are appropriate and in context. However, there could be more references to statistical models concerning MOI. While Hill and Babiker 1995 and Schneider and Escalante 2014 are appropriate, there are more statistical methods to estimate MOI, e.g., Hastings and Smith 2008 Malaria J. Particularly, COIL (Galinski et al 2015, Malaria J) and McCOIL (Chang et al, 2013, PLoS Comp. biol.) should be mentioned. A good summary is found in Schneider 2018 (PLoS One, https://doi.org/10.1371/journal.pone.0194148).


The article structure and quality of the figures are appropriate. The raw data is shared on a repository, including the R scripts used.

The article is self-contained, however the hypothesis itself needs some adjustments (see detailed comments).

Experimental design

The ms fits within the scope of the journal.

The research question per se is important and interesting. However, it is not completely meaningful. See specific comments.

The methodology is sound, and with the information provided at repositories the results are replicable.

Validity of the findings

The investigation is sufficiently novel. All underlying data is provided at a repository, and the methodology is statistically sound.
The conclusions are not always clear, which can be fixed.

Additional comments

The author studies empirically the distribution of multiplicity of infection (MOI) in malaria. MOI is recognized to be an important quantity in malaria which correlates with transmission intensity and is believed to interfere with disease severity.
Claiming that statistical methods to estimate MOI usually assume that MOI follows a Poisson or negative binomial distribution, the author fits these distributions to a dataset of Plasmodium mexicanum. The author shows that the mentioned distributions do not fit the empirical data well, while the zero-inflated Poisson and negative binomial distributions do fit well. The author further shows that there is no apparent difference between MOI in areas of low and high transmission.
While the topic itself is important I have a number of points of criticism, which require changes in the manuscript.

General comments:
1. The definition of MOI in the literature is ambiguous. In the empirical literature, it is usually defined as the number of distinct “haplotypes” in an infection. In the theoretical literature, it is defined as the number of haplotypes (not necessarily different) infecting a host, which is well summarized in Table 1. In the empirical literature, MOI is fuzzily defined: namely, each parasite is an individual and hence its own haplotype, although what is meant is actually the number of distinct haplotypes (characterized by a handful of genetic markers) that infected the host. This is then often estimated at the maximum number of alleles found in a sample from the infected host across several markers. Sometimes it is also defined as the average number of alleles across several markers. In the theoretical literature, MOI is an unobservable quantity that allows being infected multiple times with the same parasite haplotype. These ambiguities in the definition of MOI should be discussed carefully in the introduction.

2. In the manuscript, it is claimed that MOI is assumed to follow a negative binomial or Poisson distribution in the theoretical literature. While this is true, the article uses a different definition of MOI, namely the maximum number of alleles across several markers. Even if MOI follows a Poisson or negative binomial distribution, the maximum number of distinct alleles across several markers will no longer follow these distributions - and there is no statistical model available that would claim so. This is the logical mistake made in this article. In a revision, this should be discussed carefully. Furthermore, rather than fitting Poisson or negative binomial distribution to the empirical data, MOI has to be derived by a statistical model along with allele/haplotype frequencies. From these estimates, the distribution of the maximum number of alleles across the considered markers has to be derived and compared to the empirical data. These issues MUST be resolved in a revised version of the article. Otherwise, the results just state that empirical data does not fit a distribution, which no statistical model claims it does.

3. In practice, in human malaria, where MOI is considered to be important, typically (as claimed in the ms) only data from infected hosts is available. Hence, MOI (given samples descend from infected hosts) follows a conditional or zero-truncated distribution. What the article does not make clear explicitly is that conditioning the Poisson and zero-inflated Poisson (or negative binomial and zero-inflated negative binomial distribution) yields the same conditional distribution. Considering this, in the manuscript, it is not made clear why it is important to consider an unconditional model. While the article shows that the zero-inflated distribution fits empirical data well when also considering uninfected hosts, it actually gives evidence that assuming a conditional Poisson or negative binomial model, when considering only infected hosts, is appropriate. While this is stated in the conclusions it is not made as explicit in the abstract and introduction.

4. A problem that arises in the study design is the use of Plasmodium mexicanum data. In humans, MOI as the number of super- or co-infections during one disease episode makes sense, as infections are cured by chemotherapy. However, in lizards, malaria will be chronic. Therefore, MOI is the number of parasites that infect the lizard from birth to the time of sample collection. This is very different. Intuitively, MOI should hence be higher even in areas of low transmission in P. mexicanum, in agreement with the fact that MOI does not significantly differ between low and high transmission areas. This needs to be addressed in more detail.

5. I found the method section starting from L183 rather confusing and needed to go over it several times to understand the study design. Particularly, it was unclear to me which samples were analyzed for which type of analysis.

Specific comments:

1. The methods are not always clear. For instance, fitting the Poisson distribution with the function glm() in R. The GLM used here is actually a Poisson regression with just an intercept kept in the design matrix. Thus, fitting a glm() is just equivalent to ML fit of the observed count data to the desired distributions. In the case of the Poisson distribution, the Poisson parameter is estimated to be the sample size, divided by the number of events. This should be made clear explicitly.

---

## Round 0.3 · accepted · Accept

· Academic Editor

Accept

I would like to congratulate the author for this interesting work. The author has satisfactorily responded to all reviewer's questions and made the necessary changes to the manuscript.